

# Proficient brain for optimal performance: the MAP model perspective

Maurizio Bertollo[1,2], Selenia di Fronso[1,2], Edson Filho[1,3], Silvia Conforto[4], Maurizio Schmid[4], Laura Bortoli[1,2], Silvia Comani[1,5] and Claudio Robazza[1,2]

[1] BIND—Behavioral Imaging and Neural Dynamics Center, University "G. d'Annunzio" Chieti-Pescara, Chieti, Italy
[2] Department of Medicine and Aging Sciences, University "G. d'Annunzio" Chieti-Pescara, Chieti, Italy
[3] School of Psychology, University of Central Lancashire, Preston, Lancashire, United Kingdom
[4] Department of Engineering, Roma Tre University, Rome, Italy
[5] Department of Neuroscience, Imaging and Clinical Sciences, University "G. d'Annunzio" Chieti-Pescara, Chieti, Italy

Corresponding author
Maurizio Bertollo,
m.bertollo@unich.it

## ABSTRACT

**Background.** The main goal of the present study was to explore theta and alpha event-related desynchronization/synchronization (ERD/ERS) activity during shooting performance. We adopted the idiosyncratic framework of the multi-action plan (MAP) model to investigate different processing modes underpinning four types of performance. In particular, we were interested in examining the neural activity associated with optimal-automated (Type 1) and optimal-controlled (Type 2) performances.

**Methods.** Ten elite shooters (6 male and 4 female) with extensive international experience participated in the study. ERD/ERS analysis was used to investigate cortical dynamics during performance. A $4 \times 3$ (performance types $\times$ time) repeated measures analysis of variance was performed to test the differences among the four types of performance during the three seconds preceding the shots for theta, low alpha, and high alpha frequency bands. The dependent variables were the ERD/ERS percentages in each frequency band (i.e., theta, low alpha, high alpha) for each electrode site across the scalp. This analysis was conducted on 120 shots for each participant in three different frequency bands and the individual data were then averaged.

**Results.** We found ERS to be mainly associated with optimal-automatic performance, in agreement with the "neural efficiency hypothesis." We also observed more ERD as related to optimal-controlled performance in conditions of "neural adaptability" and proficient use of cortical resources.

**Discussion.** These findings are congruent with the MAP conceptualization of four performance states, in which unique psychophysiological states underlie distinct performance-related experiences. From an applied point of view, our findings suggest that the MAP model can be used as a framework to develop performance enhancement strategies based on cognitive and neurofeedback techniques.

## INTRODUCTION

This article is based on the distinction between performance effectiveness and processing efficiency (*Eysenck & Calvo, 1992*; *Eysenck et al., 2007*), and concerns the functioning of athletes during different types of optimal and suboptimal performance as conceived in

the multi-action plan (MAP) model (*Bortoli et al., 2012*). Performance effectiveness has been defined as the quality of performance (e.g., the shooting scores). Processing efficiency regards the relationship between performance effectiveness and the use of resources or effort. According to *Eysenck & Derakshan (2011)*, "…processing efficiency is high when performance effectiveness is high and use of resources is low and it is low when performance effectiveness is low but use of resources is high" (p. 956). However, this definition is not always tenable. Applied and theoretical studies observing the interaction between performance effectiveness and processing efficiency have shown that elite athletes can function in different modes when experiencing optimal performance (*Bortoli et al., 2012*; *Bertollo et al., 2013*; *Carson & Collins, 2016*; *Furley, Schweizer & Bertrams, 2015*; *Swann et al., 2016*). Athletes are able to attain effective performances not only when they execute in automatic, fluent, and flow-like states, but also when they proficiently exert some effort to cope with distressful situations. Proficient athletes, indeed, are able to achieve performance goals or results with a high degree of certainty, minimum energy expenditure, and minimum movement time in different states and conditions (*Magill & Anderson, 2014*). Proficiency is an individual's capability of achieving a particular and complex task by switching effectively between an automated and a more controlled execution according to the task and situational demands.

According to this view, two qualitatively different modes of processing, namely, automatic and controlled, have been recently proposed in the sport science field drawing on existing theories in general psychology (see *Furley, Schweizer & Bertrams, 2015*). Both types of performance are optimal, but Type 1 (fluent, automatic, and procedural) autonomous processing would be initiated and completed in the presence of relevant triggering conditions, whereas Type 2 (competent, regulated, and declarative) controlled processing would rely on working memory capacity to deal with novel problems, uncommon situations or unexpected events.

Referring to the performance effectiveness and processing efficiency distinction, optimal performance can be achieved in conditions of either efficient processing (i.e., fluent performance and low attentional control and/or effort) or effortful processing (i.e., high attentional control and/or effort). In others words, proficient athletes can complete the task successfully using both efficient and effortful types of processing (*Wilson, 2008*). In the sport domain, flow-like states in which athletes perform fluently are rare. Thus, athletes need to use their proficiency, developed through a large amount of deliberate practice (*Ericsson, 2007*), to maintain high performance levels and cope with distressful situations.

## Behavioural studies on processing efficiency

The interplay between performance effectiveness and processing efficiency in elite performance has been investigated using distractions theories and self-focus theories to explain the anxiety-performance relationship (for a discussion, see *Hill et al., 2010*).

Among the distractions theories (*Eysenck & Calvo, 1992*; *Eysenck et al., 2007*), attentional control theory (ACT) is concerned with attentional control in the context of anxiety and cognitive performance (*Eysenck et al., 2007*). It has also been applied to the motor behaviour

and sport field to investigate how anxiety influences attention and performance in goal-directed motor tasks (*Coombes et al., 2009*). Evidence from the literature suggests that anxiety impairs processing efficiency more than performance effectiveness (*Coombes et al., 2009*). Furthermore, a high level of state anxiety can impair both processing efficiency and motor performance (e.g., *Wilson, Wood & Vine, 2009*). In ACT, the crucial hypothesis is that anxiety will typically impair processing efficiency more than performance effectiveness (*Eysenck & Derakshan, 2011*). Taking a different perspective, self-focus theories (explicit monitoring, *Beilock & Carr, 2001*, and reinvestment, *Masters & Maxwell, 2008*) centre on the excessive reinvestment of attention on movement execution. This conscious control of the technique through attention 'reinvestment' leads to eventual breakdown in performance and/or chocking under pressure.

Recently, *Carson & Collins (2016)* have distinguished between positive and negative self-foci in attention control, and argued that *what* and *how* performers direct their attention is crucial to performance. This view concurs with the $2 \times 2$ conceptualization in the MAP model (*Bortoli et al., 2012*), in which performance is classified in terms of both performance level (optimal and suboptimal) and action control (automatized and controlled), as explained in greater detail below.

## Brain studies on processing efficiency

In addition to the many behavioural studies reported in the literature, which support the hypothesis that anxiety could impair processing efficiency more than performance effectiveness (for a review, see *Eysenck et al., 2007*), there are several studies that have analysed processing efficiency by using different techniques for brain activity assessment (*Bishop, 2009*; *Righi, Mecacci & Viggiano, 2009*; *Savostyanov et al., 2009*). Findings showed that high levels of anxiety can be associated with greater brain activity compared to low levels, even when anxiety has no impact on performance. Using the same psychophysiological and methodological perspective, the present study focuses on the neural efficiency hypothesis as a framework to test the relationship between performance effectiveness and the use of resources or effort during elite performance in sport.

The neural efficiency hypothesis of psychomotor performance (see *Del Percio et al., 2008*; *Hatfield & Kerick, 2007*) derives from studies on the relationship between brain and intelligence, championed by Haier (*Haier et al., 1988*; *Haier et al., 1992*; *Haier et al., 2004*). Efficiency occurs as a result of the disengagement of brain areas that are irrelevant for a given task, along with the simultaneous engagement of highly task-relevant areas (*Haier et al., 1992*). In fact, extant research supports the notion that EEG topographical oscillations in the theta and alpha band are associated with cognitive performance (*Klimesch, 1999*; *Klimesch, 2012*). Overall, synchronization (ERS) in the theta band has been linked to encoding of new information in the episodic memory due to a task-related power increase, while the frontal midline theta has been associated with sustained attention and top-down processing in precision sports (*Chuang, Huang & Hung, 2013*; *Doppelmayr, Finkenzeller & Sauseng, 2008*; *Klimesch, 1999*). De-synchronization (ERD) in the lower alpha band (i.e., 8–10 Hz) is thought to reflect general task demands and attentional processes (i.e., vigilance, arousal), whereas desynchronization in the upper alpha band

(i.e., 10–12 Hz) has been associated with semantic performance (*Klimesch, 1999*) and task-related attention (*Klimesch, 2012*). Moreover, alpha ERS and ERD reflect inhibition and the release from inhibition, respectively (*Klimesch, 2012*). In addition, excessive controlled processing impairs automaticity (*Masters & Maxwell, 2008*), and is linked to higher cortical activity in the attentional network, especially in the frontal midline and parietal areas (*Kao, Huang & Hung, 2013*).

In the sport sciences domain, neural efficiency has been paralleled to "the most efficient movement" in terms of the energy cost or work output of a given movement, and has been defined as psychomotor efficiency (*Hatfield & Kerick, 2007*). High-skilled athletes usually perform with minimal effort in comparison to novices. Noteworthy, *Hatfield & Kerick (2007)* highlighted that economy of effort is a marker of superior psychomotor performance, thereby supporting the notion that high-level performance is marked by economy of brain activity with an inverse relationship between performance effectiveness and resources utilization.

Additional research on the neural efficiency hypothesis has corroborated the notion that skilled performance in self-paced sports is usually accompanied by a lower and/or task-synchronized cortical activity (*Del Percio et al., 2009a*; *Del Percio et al., 2009b*; *Del Percio et al., 2010*; *Del Percio et al., 2008*). For instance, Babiloni and colleagues (*2010*) found greater resting alpha power in elite athletes compared to amateur athletes and non-athletes. They concluded that athletes' brains are more inhibited during resting states in agreement with the neural efficiency hypothesis. Also congruent with the neural efficiency hypothesis, findings from a study by Del Percio and colleagues (*2009* and *2011*) suggest that elite athletes exhibit less alpha power reactivity than non-athletes.

Altogether, neural efficiency seems to reflect two different processes in distinguishing elite from expert and novice athletes. "The first is a reduction in neural activity in certain brain regions as a particular skill becomes more automated and less controlled. The second is a reduction of activity in sensory and motor cortex, reflecting more efficient processing made possible by less energy expenditure" (*Callan & Naito, 2014*, p. 183). Specifically, sensorimotor rhythms (12–15 Hz) are involved in visuo-motor tasks and may represent adaptive information processing during motor execution (*Cheng et al., 2015a*), sensitive to neurofeedback intervention (*Cheng et al., 2015b*).

Although a number of studies support the neural efficiency hypothesis, there is emerging evidence suggesting that this hypothesis does not fully account for elite athletes' brain activity in self-paced sports (*Vecchio, Del Percio & Babiloni, 2012*). In particular, two recent experimental studies in golf did not lend support to the neural efficiency hypothesis. In the first study, Babiloni and colleagues showed that alpha ERD during complex visuo-motor integration was higher when comparing athletes with non-athletes, and best performance with worst performance in elite athletes (*Babiloni et al., 2008*). In the second study, they observed that successful putts of elite golfers were related to strong parieto-central and parieto-frontal alpha coherence (*Babiloni et al., 2011*).

Of note, the neural efficiency hypothesis emerged mainly from studies comparing experts and novices (i.e., expert-novice paradigm), and thus might not illustrate the unique neural mechanisms associated with expert performance in sports (expert-performance

approach), in the sense that it does not explain the interplay between processing efficiency and performance effectiveness. Therefore, the unique neural mechanisms associated with optimal and suboptimal performance states among skilled athletes remain an understudied topic (*Comani et al., 2014*; *Bertollo et al., 2013*; *Di Fronso et al., 2016*). Put differently, it is established that experts will show a greater degree of automaticity and neural efficiency when compared to novices (*Callan & Naito, 2014*; *Del Percio et al., 2008*). However, it remains mostly unknown whether and how neural efficiency among brain cortices is related to different performance states among highly skilled athletes.

## The MAP model

To advance the knowledge on the unique psychophysiological and behavioural mechanisms underlying optimal and suboptimal performance states among high-skilled athletes, Bortoli and colleagues (*2012*) have developed the MAP intervention model. The model is embedded within the IZOF framework (*Hanin, 2007*), and fits well with the ACT hypothesis that anxiety will typically impair processing efficiency more than performance effectiveness. It also concurs with dual-process theories that describe and explain the dual modes of processing of athletes, namely, automatic and controlled (see *Furley, Schweizer & Bertrams, 2015*). In this view, Type 1 automatic processing allows for the fast and effortless execution of behavioural responses, whereas Type 2 controlled processing is well equipped for dealing with novel problems or unexpected events. Furthermore, the MAP model is rooted in a motoric perspective, in which skill establishment is related to aspects governing motor control. From this viewpoint, skill establishment concerns the structures/representations of the movement and refers to the level and consistency of automaticity in the execution of different movement components (*Carson & Collins, 2016*).

According to the MAP model (*Bortoli et al., 2012*), performance experience can be depicted along two different axes: one represents the performance level per se, and the other represents the amount of control over the task. According to this interaction, this performance experience plane can be classified into four performance Types: optimal-automatic (Type 1), optimal-controlled (Type 2), suboptimal-controlled (Type 3), and suboptimal-automatic (Type 4; see Fig. 1). In Type 1 performance (flow condition) the athlete is self-confident and in control of the situation, and is able to direct his or her physical and mental energies toward the accomplishment of a task. This type of processing is independent of available working memory capacity. The movement is effortless, autonomous, and consistent and attentional control systems play a role of supervision. Both performance effectiveness and processing efficiency are high. Stress, fatigue, unexpected events, or performance issues could disrupt performance. These unfavourable conditions can lead to Type 3 performance, typified by an excessive reinvestment in controlled processing and an overload of working memory, which undermines movement fluidity and automaticity (*Maxwell, Masters & Eves, 2000*; *Oudejans et al., 2011*). Both performance effectiveness and processing efficiency are low. Finally, Type 2 performance state is characterized by high performance effectiveness and low processing efficiency (i.e., effortful processing and involvement of working memory; *Eysenck & Derakshan, 2011*), while Type

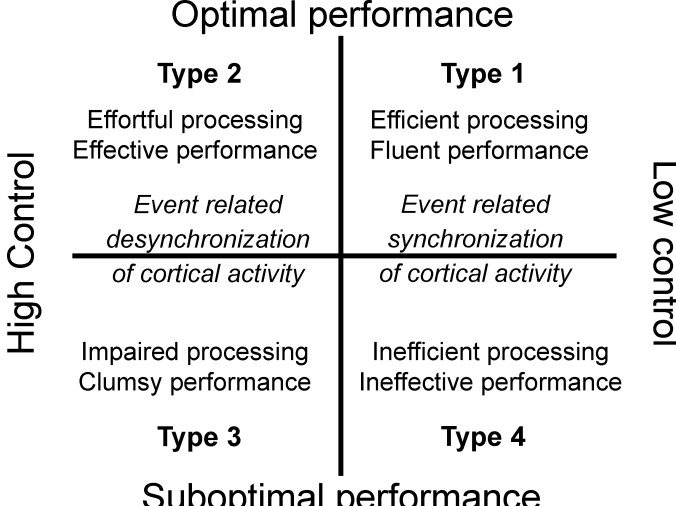

**Figure 1** The multi-action plan model.

4 performance is typified by ineffective performance, inefficient processing, and no reliance on working memory.

Research findings on the MAP model have shown that different physiological (e.g., skin conductance responses, breathing rate, and heart rate) and behavioural markers (e.g., kinematic patterns) underlie the four types of performance states (*Bertollo et al., 2013*; *Filho et al., 2015*) and focus of attention can affect performance (*Bertollo et al., 2015*). There is also initial evidence that different neural patterns are associated with the four performance types foreseen in the MAP model (*Comani et al., 2014*; *Di Fronso et al., 2016*). Specifically, *Comani et al. (2014)* observed that optimal-automatic performance (i.e., Type 1) states among shooters were characterized by lower alpha power in the central, contralateral parietal, and occipital areas (at shot release), in agreement with the neural efficiency hypothesis (i.e., global decrease in cortical activity). Conversely, optimal-controlled performance (i.e., Type 2) was characterized by an increased alpha power activity in the frontal and occipital areas.

## The present study

In the present study, we aimed to further explore the neural dynamics associated with optimal/suboptimal by controlled/automated performance states among high-skilled athletes in accordance with the MAP model tenets. We were particularly interested in examining the neural activity associated with optimal-automated (Type 1) and optimal-controlled (Type 2) performances in the light of the reviewed literature on ACT theory, neural efficiency, and dual-process theories in sport and motor performance. According to the MAP model assumptions and previous research findings, we expected to find different patterns of cortical activity associated with the four types of performance states. In particular, we expected to observe more cortical synchronization in Type 1 and Type 4 performance states (see Fig. 1). This hypothesis relies on earlier findings that showed some similar psychophysiological features underlying Type 1 and Type 4 performances

(*Bertollo et al., 2013*; *Di Fronso et al., 2016*; *Robazza et al., in press*). On the other hand, we expected to observe more cortical desynchronization associated with Type 2 and Type 3 performance states. Specifically, we expected to find the following cortical patterns:

(1) Type 1 performance: (a) greater ERS in the theta band reflecting the successful encoding of information and the activity of the attentional network; (b) at the same time, ERS activity in both low and high alpha band, which is an expression of the inhibition processes (*Klimesch, 2012*) and neural efficiency (*Del Percio et al., 2008*).

(2) Type 2 performance: (a) an ERS activity similar to Type 1 performance, with more ERD in the low and high alpha band, reflecting the two faces of inhibition, namely, selective activation vs. blocking of information processing; (b) more selective activation related to the attentional control exerted on the action.

(3) Type 3 performance: (a) a clear ERD pattern in the theta band, indicating specific attention to the task at hand. Indeed, theta response over frontal regions reflects the activation of neural networks involved in the allocation of attention related to target stimuli: a parietal activation would occur during the processing of visuospatial information, and a premotor cortex activation would occur during visuo-motor processing; (b) more ERD in the low alpha band associated with higher spread attention engaged and effort resources utilization; (c) more ERD in the areas associated with verbal semantic processes or visuo-spatial processing.

(4) Type 4 performance: (a) a cortical activity pattern similar to Type 1 performance; (b) a hyper-synchronization (high ERS) during the entire interval preceding execution because of low controlled and disengaged behaviour with no reliance on working memory. It is well known that hyper-synchronization is associated with no reliance on working memory and a sort of loss of consciousness derived from task disengagement (*Klimesch, 2012*).

## METHOD

### Participants

Ten elite, right-handed shooters (6 male and 4 female), with extensive experience in international competitions (e.g., World Championships, World Cup Championships, Olympic Games) and included in the International Shooting Sport Federation international ranking, participated in the study. The participants ranged in age from 18 to 29 years ($M = 22.8$, $SD = 3.5$) and had at least 10 years of experience ($M = 14.5$, $SD = 4.0$). After being briefed on the general purpose of the study, the participants agreed to participate and signed a written informed consent. The study was conducted in accordance with the declaration of Helsinki and received approval from the local university ethics committee (ref. n. 10-21/05/2015).

### Procedure

The participants were asked to identify the core components of their "chain of action" that were essential for optimal performance. They were then asked to choose one element (i.e., an idiosyncratic core component, such as aiming, triggering, and front sight alignment) deemed fundamental in order to perform optimally according to the procedure developed

by *Bortoli et al. (2012)*. Subsequently, the participants performed a total of 120 air-pistol shots. They were "free to relax" between shots and to shoot when they felt "ready to go" (average inter-shot interval of about 1 min). The distance between the shooter and the target was 10 m, and the diameter of the target was 6 cm, in accordance with international shooting competition rules (www.issf-sports.org/theissf/rules/english_rulebook.ashx). An electronic scoring target (Hs10 Hybridscore, SIUS, Effretikon, Switzerland) was used to (automatically) record the shooting scores. Shooting score results were initially concealed from the athletes as we were interested in assessing their perceived accuracy. Hence, after each shot, the athletes were asked to report their perceived shooting score (ranging from 0 to 10.9). They also reported the control level of the idiosyncratic core component of action (from 0 to 11 on a Borg scale; for a detailed description see *Bertollo et al., 2013*). After this evaluation, the actual shooting score (i.e., objective performance) of each shot was made available to the shooter. In addition to self-evaluation, we also recorded their brain waves using a 32-channel waveguard cap by ANT (Advanced Neuro Technology, Enshede, The Netherlands).

### EEG recordings

EEG data were continuously recorded (sampling frequency: 1024 Hz) from the 32 scalp electrodes (active electrodes for movement compensation) positioned over the scalp according to the 10–20 system, using an eegosport$^{TM}$ amplifier (ANT, Enschede, Netherlands). EEG signals were recorded with common reference. The ground electrode was positioned between Fpz and Fz, and the electrode impedance was kept below 10 kΩ. A device based on acoustic technology (cardio-microphone and Powerlab 16/30, ADInstruments, Australia) was used to identify the instant of shot release. It was positioned on a tripod 30 cm from the air pistol and the acoustic signals were acquired with a sampling frequency of 1 kHz.

### Performance categorization

The participant' shooting scores and perceived control levels were used to categorize the EEG epochs into a 2 × 2 matrix using the median split technique to identify the four types of performance as defined in the MAP model. Following this approach, shooting results from 10.2 to 10.9 (maximal possible score) were categorized as optimal, and the remaining scores as suboptimal. Attentional control levels ≤ 4 were categorized as automatic performance and the others as controlled performance. Therefore, the combination of shooting result > 10.2 and control level ≤4 led to a performance categorization as optimal-automatic, i.e., as Type 1, whereas the performance was classified as optimal-controlled, i.e., Type 2 with the combination of shooting result > 10.2 and control level > 4. When the shooting result was <10.2 and the control level was >4, performance was considered as suboptimal-controlled, i.e., Type 3, whereas performance was classified as suboptimal-automatic, i.e., Type 4 when the shooting result was <10.2 and the control level was ≤4.

### EEG data pre-processing

EEG data were band-pass filtered between 0.3 and 40 Hz, and segmented into single epochs of 10 s duration. Each epoch started at −6 s and ended at +4 s with respect to shot release

($t = 0$). Data epochs were visually inspected, and those showing instrumental, ocular and muscular artefacts were corrected using the tool available in the ASA software (*Zanow & Knosche, 2004*). The data epochs showing residual artefacts were excluded from further analysis.

### ERD/ERS analysis

We were particularly interested in the relationship between attention and performance. Thus, we ran the ERD/ERS analysis in the theta, and low and high alpha frequency bands. To quantify the event-related changes in these frequency bands, the individual ERD/ERS maps were calculated following the procedure proposed by *Zanow & Knosche (2004)*, where ERD and ERS are defined, respectively, as the percent increase and decrease of signal power as compared to the baseline. This definition is different from that proposed by *Pfurtscheller & Lopes da Silva (1999)*. Following the procedure proposed by *Zanow & Knosche (2004)*, the Hilbert transform was performed before ERD/ERS analysis. For a given frequency band, ERD/ERS maps were calculated for an interval of interest as the percent variation of the signal power with respect to baseline values for each EEG channel.

Given that the individual alpha frequency was similar across participants (9.9–10.1 Hz), we considered the following frequency bands: 4–8 Hz (i.e., theta), 8–10 Hz (low alpha), and 10–12 Hz (high alpha) for analysis. Three intervals of interest, each of 1 s duration, were considered during the 3 s preceding each shot (i.e., from −3 to 0), whereas the baseline signal was windowed from −5 to −4 s before the shot. Periods before −5 s were not suitable for analysis because of body movements, small adjustments of the head/trunk, and respiration. For each participant and for each group of EEG epochs categorized according to the types of performance foreseen in the MAP model, the baseline signals were averaged to reduce background noise before ERD/ERS calculation. Similarly, averaged ERD/ERS maps were obtained from the single ERD/ERS maps for each interval of interest, for each participant and for each group of EEG epochs. Finally, for each type of performance the individual ERD/ERS maps were averaged across subjects to account for the cortical patterns underlying the four MAP model types, thus obtaining average topographical maps.

### Statistical analysis

The percentage of ERD/ERS individual data was exported with ASA software (ANT-Neuro) and analysed using Statistica 10 software (Statsoft). Separated repeated measures analyses of variance (RM-ANOVAs) for each electrode were applied to each frequency band (theta, low alpha, and high alpha). Huynh–Feldt correction was applied (*Huynh & Feldt, 1976*) when the assumption of sphericity was violated.

## RESULTS

Descriptive statistics for the behavioural data for each performance type as derived from the interaction of optimal/suboptimal and automatic/controlled performance dimensions are summarized in Table 1.

A 4 × 3 (performance types × time) repeated measures analysis of variance (RM-ANOVA) was performed to test the differences among the four types of performance

**Table 1  Mean and standard deviation during each performance type of shooting result and control level on the core component of the action.**

| Performance type | Number of shots[a] | Shooting results | Control level on core components |
|---|---|---|---|
| 1 | 238 | 10.51 (0.12) | 4.59 (0.39) |
| 2 | 350 | 10.45 (0.12) | 6.47 (0.54) |
| 3 | 212 | 9.90 (0.19) | 6.44 (0.46) |
| 4 | 348 | 9.76 (0.24) | 4.39 (0.54) |

Notes.

[a]The number of shots for each performance type derives from the number of trials (120) for each athlete (10) without the shots affected by EEG artefacts.

**Table 2  Significant results of the RM-ANOVAs 4 × 3 (performance × time) on the ERD/ERS theta band.**

| | | Theta ERD/ERS | | | |
|---|---|---|---|---|---|
| Variable | Electrodes | Degrees of freedom | $F$ | $p$ | $\eta_p^2$ |
| Performance | Fpz | 3,27 | 5.338 | .005 | .372 |
| Performance | FC2 | 3,27 | 4.230 | .014 | .320 |
| Performance | CP2 | 3,27 | 4.054 | .016 | .311 |
| Performance | CP6 | 3,27 | 3.222 | .038 | .264 |
| Time | F7 | 2,18 | 5.005 | .018 | .357 |
| Time | FC1 | 2,18 | 5.317 | .015 | .371 |
| Time | FC5 | 2,18 | 7.316 | .004 | .410 |
| Time | T7 | 2,18 | 7.888 | .003 | .467 |
| Time | Cz | 2,18 | 4.054 | .035 | .311 |
| Time | CP1 | 2,18 | 8.858 | .001 | .496 |
| Time | CP5 | 2,18 | 3.972 | .037 | .306 |
| Performance × Time | Fz | 6,54 | 2.033 | .047 | .204 |
| Performance × Time | CP6 | 6,54 | 2.680 | .023 | .229 |

during the three seconds preceding the shots for theta, low alpha and high alpha frequency bands. The dependent variables were the ERD/ERS percentages in each frequency band (i.e., theta, low alpha, high alpha) for each electrode site across the scalp. This analysis was conducted on approximately 90 shots for each participant in three different frequency bands.

## Theta ERD/ERS

A series of RM-ANOVAs 4 × 3 (performance × time) on the ERD/ERS theta band yielded significant effects on performance, time, and the interaction performance by time. Only the significant results are reported in Table 2.

Figure 2 illustrates the grand averages of the scalp topographical distributions of the ERD/ERS amplitudes in the theta band for each performance type. The theta ERD/ERS was mapped during three pre-shot periods: from −3 s to −2 s, from −2 s to −1 s, and from −1 s to shot release ($t = 0$).

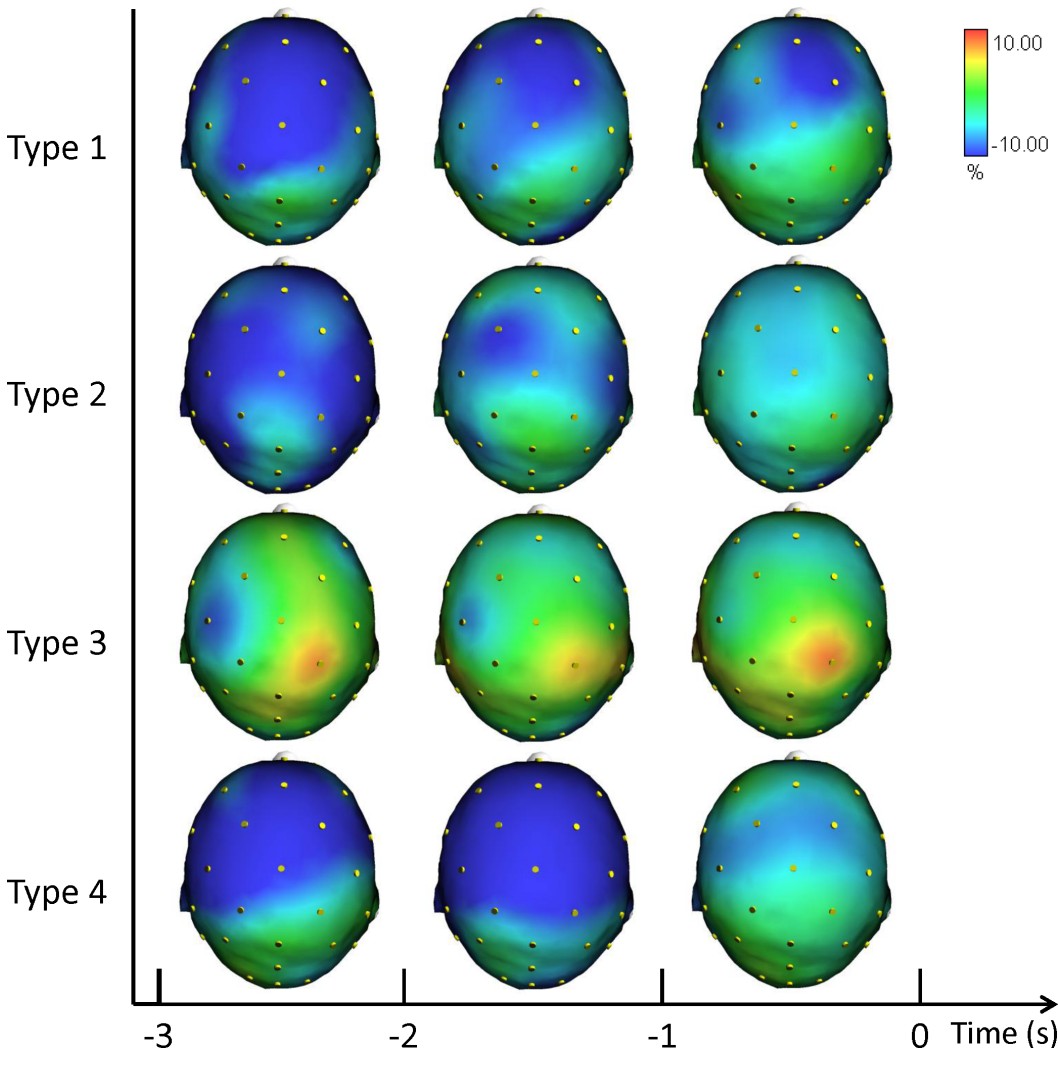

**Figure 2** **Average topographical distributions of the ERD/ERS amplitudes in the theta band for each performance type during the 3 s before shot release.** The theta ERD/ERS maps were calculated for three pre-shot periods: from −3 s to −2 s, from −2 s to −1 s, and from −1 s to shot release ($t = 0$). Time scale is on the $X$ axis. Color scale: maximum ERD and ERS are coded in red and blue, respectively. The maximal (%) value of the ERD/ERS is given at the top right side of the figure.

Given our interest in performance comparison, we performed a post-hoc analysis for performance and performance by time interaction. Fisher post-hoc analyses for the significant results in theta band are reported in Supplemental Information.

### Low alpha ERD/ERS

A series of RM-ANOVAs 4 × 3 (performance × time) on the ERD/ERS low alpha band yielded significant effects on performance and time. The significant results are reported in Table 3. No differences on the performance by time interaction were found.

In Fig. 3 we plotted the grand averages of the scalp topographical distributions of the ERD/ERS amplitudes in the low alpha frequency band for each performance type. The low alpha ERD/ERS was mapped at three pre-shot periods: from −3 s to −2 s, from −2

**Table 3** Significant results of the RM-ANOVAs 4 × 3 (performance × time) on the ERD/ERS low alpha band.

| | | Low alpha ERD/ERS | | | |
|---|---|---|---|---|---|
| Variables | Electrodes | Degrees of freedom | $F$ | $P$ | $\eta_p^2$ |
| Performance | F8 | 3,27 | 3.279 | .040 | .261 |
| Performance | FC2 | 3,27 | 5.105 | .006 | .358 |
| Time | Fp1 | 2,18 | 3.969 | .048 | .306 |
| Time | Fpz | 2,18 | 4.635 | .044 | .340 |
| Time | F7 | 2,18 | 8.905 | .002 | .497 |
| Time | F3 | 2,18 | 3.767 | .042 | .295 |
| Time | F4 | 2,18 | 8.809 | .002 | .495 |
| Time | F8 | 2,18 | 3.845 | .040 | .299 |
| Time | FC5 | 2,18 | 8.879 | .002 | .497 |
| Time | FC1 | 2,18 | 4.504 | .025 | .334 |
| Time | T7 | 2,18 | 5.544 | .028 | .381 |
| Time | C3 | 2,18 | 9.101 | .001 | .503 |
| Time | Cz | 2,18 | 6.124 | .009 | .405 |
| Time | Cp5 | 2,18 | 6.193 | .008 | .408 |
| Time | Cp1 | 2,18 | 10.926 | .001 | .548 |
| Time | Oz | 2,18 | 5.842 | .011 | .394 |
| Time | O2 | 2,18 | 5.554 | .013 | .382 |

s to −1 s, and from −1 s to shot release ($t = 0$). Similiarly to the theta band results, we performed the post hoc analysis for performance and performance by time interaction. Fisher post-hoc analyses for the significant results in low alpha band are included in Supplemental Information.

## High alpha ERD/ERS

A series of RM-ANOVAs 4 × 3 (performance × time) on the ERD/ERS high alpha band yielded significant effects on performance, time, and the performance by time interaction. The significant results are reported in Table 4.

Figure 4 shows the grand averages of the scalp topographical distributions of the high ERD/ERS amplitude in the alpha frequency band for each performance type. The high alpha ERD/ERS was mapped during three pre-shot periods: from −3 s to −2 s, from −2 s to −1 s, and from −1 s to zero time.

Similarly to the previous results, we performed a post-hoc analysis for performance and performance by time interaction. Fisher post-hoc analyses for the significant results in the high alpha band are included in Supplemental Information.

## DISCUSSION

In the present study, we used a mobile EEG device to analyse the cortical activity underlying different types of optimal and suboptimal performance states in the framework of the MAP model. Assessing brain activity using mobile EEG data collection enables evaluation of neural efficiency during performance in actual settings (*Park, Fairweather & Donaldson,*

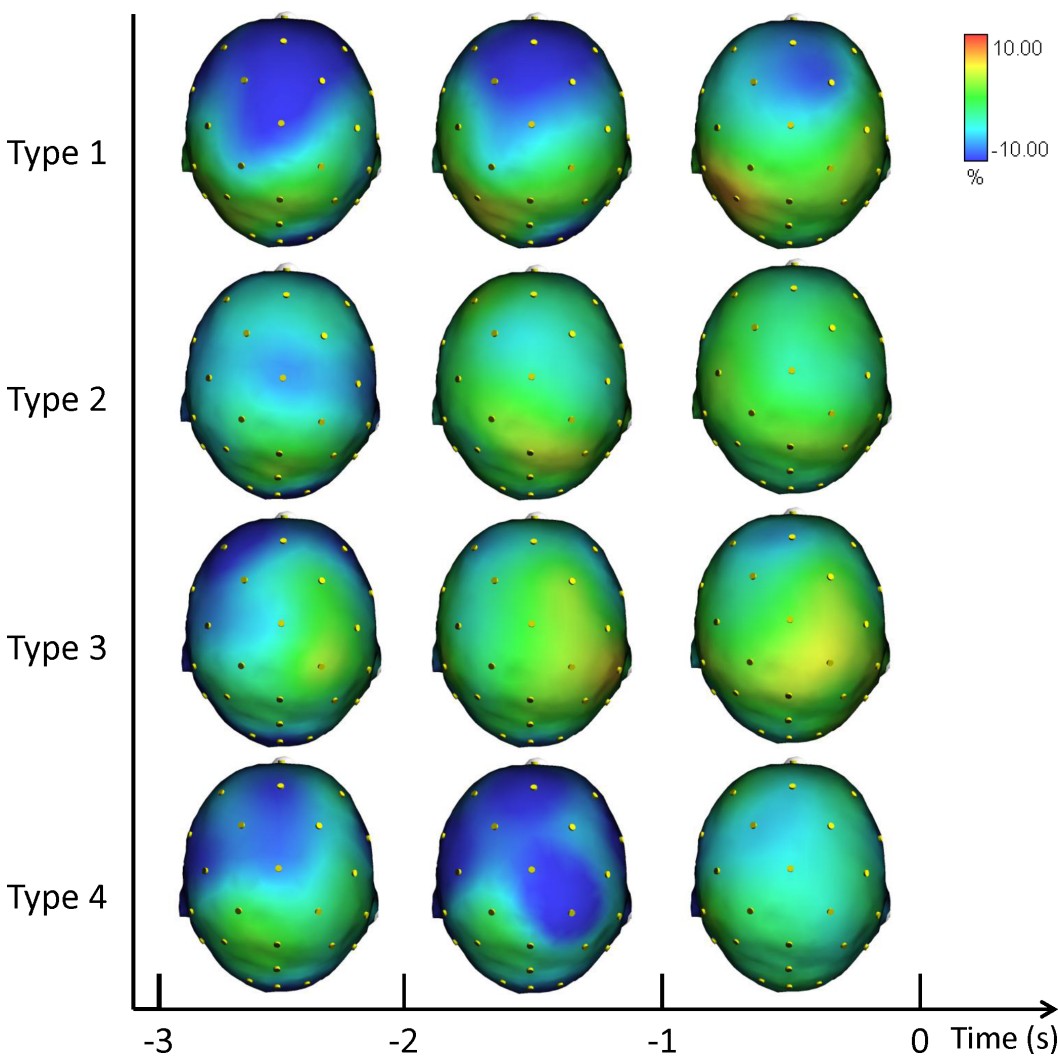

**Figure 3** **Average topographical distributions of the ERD/ERS amplitudes in the low alpha frequency band for each performance type during the 3 s before shot release.** The low alpha ERD/ERS maps were calculated for three pre-shot periods: from −3 s to −2 s, from −2 s to −1 s, and from −1 s to shot release (t = 0). Time scale is on the X axis. Colour scale: maximum ERD and ERS are coded in red and blue, respectively. The maximal (%) value of the ERD/ERS is given at the top right side of the figure.

*2015*). Our findings showed that elite athletes used different types of functioning to achieve optimal performance, sometimes typified by effort, attentional control, and use of resources higher than in flow-like automatic performance (*Bortoli et al., 2012*; *Carson & Collins, 2016*; *Furley, Schweizer & Bertrams, 2015*). This outcome is in line with the view that "the processing inefficiency caused by the disruption of the inhibition and shifting functions of the central executive system does not necessarily lead to decrements in performance effectiveness ... by using compensatory or alternative processing strategies" (*Wilson, 2008*, p. 195). Similar behaviour has been observed when people have to cope with affordances that are not always appropriate in a particular situation, and therefore need to be inhibited by higher cognitive functions located in the frontal lobes (*Furley, Schweizer*

**Table 4** Significant results of the RM-ANOVAs 4 × 3 (performance × time) within subjects on 432 the ERD/ERS high alpha band.

| | High alpha ERD/ERS | | | | |
|---|---|---|---|---|---|
| Variables | Electrodes | Degrees of freedom | $F$ | $p$ | $\eta_p^2$ |
| Performance | Fp1 | 3,27 | *3.570* | .026 | .284 |
| Performance | F8 | 3,27 | 3.038 | .046 | .252 |
| Performance | P3 | 3,27 | 3.721 | .023 | .292 |
| Time | Fp1 | 2,18 | 5.460 | .029 | .378 |
| Time | Fp2 | 2,18 | 4.811 | .040 | .348 |
| Time | F7 | 2,18 | 4.670 | .049 | .342 |
| Time | F4 | 2,18 | 8.354 | .002 | .481 |
| Time | Fc5 | 2,18 | 8.186 | .002 | .476 |
| Time | Fc2 | 2,18 | 4.846 | .020 | .350 |
| Time | T7 | 2,18 | 4.813 | .021 | .348 |
| Time | C3 | 2,18 | 4.689 | .022 | .343 |
| Time | Cz | 2,18 | 4.322 | .029 | .329 |
| Performance × Time | Fp1 | 6,54 | 2.968 | .033 | .248 |
| Performance × Time | Fz | 6,54 | 2.401 | .039 | .211 |
| Performance × Time | F8 | 6,54 | 3.196 | .009 | .262 |
| Performance × Time | C3 | 6,54 | 3.473 | .005 | .224 |
| Performance × Time | Cp1 | 6,54 | 2.401 | .039 | .278 |

*& Bertrams, 2015*). In brain studies, ERD has been associated with higher involvement of cognitive processes (in particular working memory), and reflects anticipatory attention (*Klimesch, 2012*; *Nussbaumer, Grabner & Stern, 2015*).

In the sport science domain, there is emerging evidence suggesting that the neural efficiency hypothesis does not fully account for elite athletes' brain activity (*Vecchio, Del Percio & Babiloni, 2012*). Vecchio and colleagues have argued that information processing in the cognitive-motor cortical systems of elite athletes is more complex than that predicted by the "neural efficiency" hypothesis. In experts, some neural networks within the cortex might reflect the "neural efficiency" (e.g., default mode network and relaxation), whereas other networks might reflect other mechanisms (e.g., task request, attentional demands, or affective demands). In cognitive neuroscience, *Poldrack (2015)* recently raised doubts about efficiency as a useful concept to explain expert performance. He proposed that the common usage of the concept of "efficiency" in cognitive neuroscience is equally vacuous. In general, the term is used to describe situations where performance appears similar but activation is greater for one group (e.g., novices, which are usually labelled as "less efficient").

In discussing the brain studies associated with superior performance, we adopted the idiosyncratic framework of the MAP model to test different processing modes underpinning different types of performance. In this framework, we hypothesized to integrate the neural efficiency hypothesis within a broader concept, namely, the neural proficiency hypothesis of superior performance, in which athletes' brain waves are modulated by the individual's effort to maintain a high performance level by switching proficiently between Type 1 and Type 2 performance states. Proficiency is an active and qualitative process that involves the

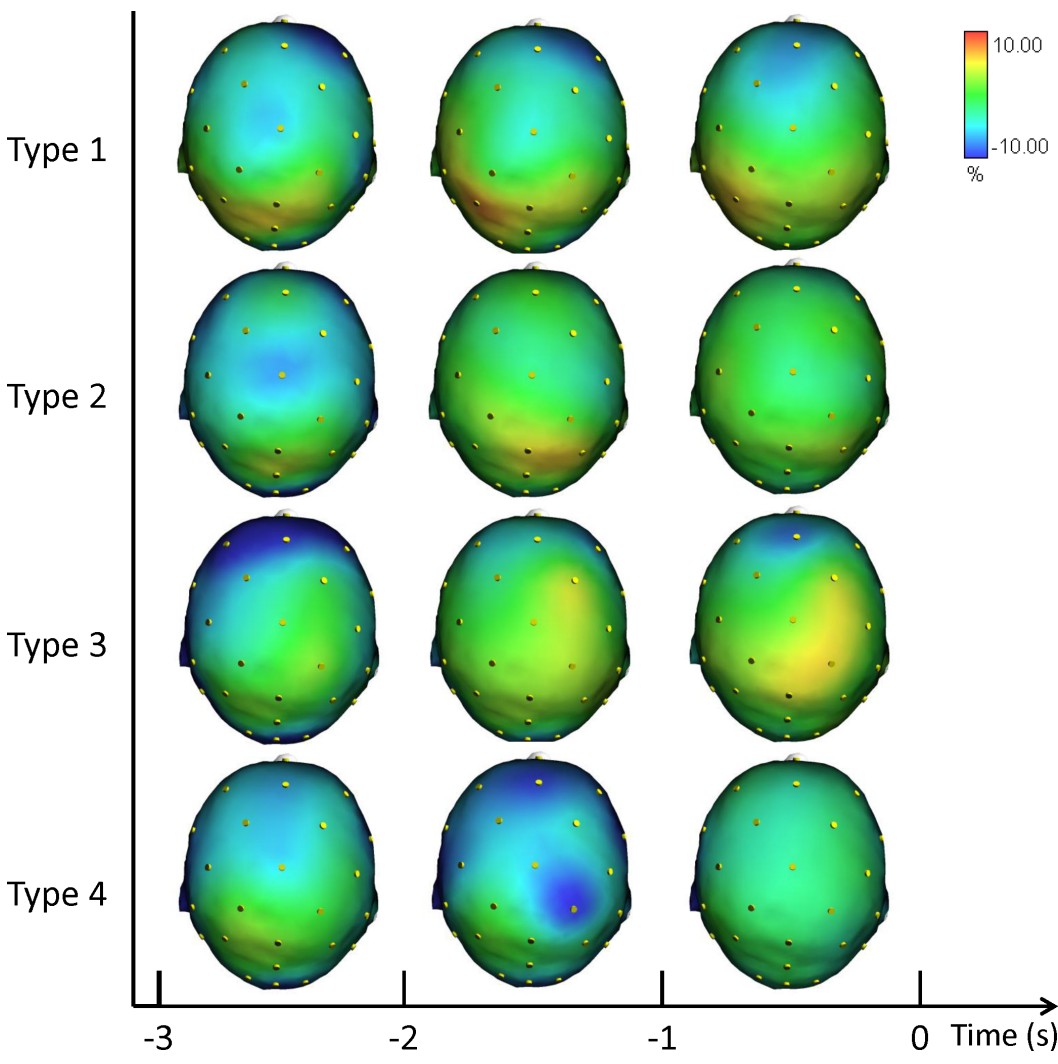

**Figure 4** **Average topographical distributions of the ERD/ERS amplitudes in the high alpha frequency band for each performance type during the 3 s before shot release.** The high alpha ERD/ERS maps were calculated for three pre-shot periods: from $-3$ s to $-2$ s, from $-2$ s to $-1$ s, and from $-1$ s to shot release ($t = 0$). Time scale is on the $X$ axis. Colour scale: maximum ERD and ERS are coded in red and blue, respectively. The maximal (%) value of the ERD/ERS is given at the top right side of the figure.

interaction between the efficiency of human processing and the efficacy of the performance, and it is not only the ratio between the input and the output of the biomechanical system to reach the outcome. A proficient brain is the ensemble of the neural features that define the state of being adequate or well qualified both physically and psychologically for the task. This implies that efficient processing and inefficient processing (measured through cortical activity) during performance are modulated by the degree of effort and by the attentional demands of the control systems involved in the task. This is mirrored in the neural activity found in optimal (Type 1 and Type 2) and suboptimal (Type 3 and Type 4) performance states, as found in earlier preliminary case studies with elite athletes (*Comani et al., 2014*; *Di Fronso et al., 2016*). Our findings are also in line with a study on

movement-related cortical activity as a function of expertise and pressure (*Cooke et al., 2014*), in which a greater reduction in EEG high-alpha power during preparation for action was deemed to reflect more resources being devoted to response programming. Indeed, both cortical excitatory and inhibitory factors can contribute to elite skilled performance during different types of good and poor performances. This most likely depends on the different degree of attentional requests on the task and the high anxiety levels usually found in competition that can undermine automaticity (*Hatfield et al., 2013*). In this situation, a Type 2 processing, well equipped for dealing with novel problems, is more appropriate (*Furley, Schweizer & Bertrams, 2015*).

In agreement with prior studies demonstrating that the event-related changes of the cortical desynchronization/synchronization (ERD/ERS) can capture differences in the brain activity of elite athletes during optimal and suboptimal performances (*Comani et al., 2014*; *Del Percio et al., 2009a*; *Del Percio et al., 2009b*; *Di Fronso et al., 2016*). In the present study we investigated the event-related changes of cortical activity (ERD/ERS) in the theta and alpha bands in association with different performance states. ERS gathers a similar state of neuronal assemblies resulting in summation of postsynaptic potentials (synchrony) due to similarity of neuronal states. Conversely, a dissimilar state of neuronal assemblies results in differential assignment of neurons and de-synchrony or changing power or amplitude (ERD). These two types of brain configurations are an expression of the two classical types of performance: good and poor (*Cooke, 2013*; *Del Percio et al., 2009a*; *Del Percio et al., 2009b*; *Hatfield & Kerick, 2007*). In effect, the cortical patterns displayed in Figs. 2, 3 and 4 confirm that neural desynchronization (i.e., ERD) is mainly associated with Type 3 performance (suboptimal-controlled). However, ERS activity was not only associated with optimal-automated performance (Type 1), but also associated with suboptimal-automated performance (Type 4). In addition, during optimal-controlled performance (Type 2) cortical activity showed greater ERD activity in different areas and frequency bands compared to optimal-automated performance, contrary to the neural efficiency hypothesis, which states that more synchronization would appear in association with good performance. However, when we observed a more diffused and evident ERD activity in the brain (i.e., when athletes were unable to maintain a consistent execution associated with economy of effort), we simultaneously observed a decline toward a suboptimal-controlled performance (Type 3). This pattern is more evident in the theta band and it involves the right parietal areas, in agreement with the evidence that (1) theta activity is also present at parietal sites in the high- vs. low-error condition (*Arrighi et al., 2016*), and that (2) integration of visual and spatial information involves both frontal and parietal areas (*Morgan et al., 2013*).

The ability to achieve performance goals with a high degree of certainty, and not only minimum energy expenditure, is paramount for high performance in sport. Thus, athletes need to use different strategies to maintain or achieve good performance (*Bortoli et al., 2012*; *Furley, Schweizer & Bertrams, 2015*; *Swann et al., 2016*). These strategies, that may involve the engagement of working memory, require more effort than during optimal-automatic performance experiences (Type 1). Processing efficiency tends to decrease during this type of effective performance (Type 2). Indeed, Type 2 performance of our shooters was typified

by a dissimilar state of neuronal assemblies during active task engagement, resulting in differential assignment of neurons and desynchronized theta/alpha power. Specifically, we found a significant desynchronization in the visuo-spatial and visuo-motor attentional networks (Fpz Fc2 Cp2 Cp6) in the theta band during Type 3 performance, whereas the other types of performance were still synchronized. These findings also suggest that more sustained attention on the target associated with top-down processes was necessary during execution (*Kao, Huang & Hung, 2013*; *Missonnier et al., 2006*). This is apparent from the results of the performance by time interaction in which we found significant ERD in the frontal midline theta (Fz) during the second before shot release as well as de-synchronization in parietal areas (Cp6) two seconds before shot release, consistent with findings from other studies (*Chuang, Huang & Hung, 2013*; *Doppelmayr, Finkenzeller & Sauseng, 2008*; *Kao, Huang & Hung, 2013*).

Previous studies have found that neural networks engaged in the control of action are dynamically reorganized depending on whether the task calls for volitional control or is performed automatically (see *Matsuzaka, Akiyama & Mushiake, 2013*). Specifically, the prefrontal cortex and anterior striatum were shown to exhibit task-related activity modulation (i.e., synchronization) under the former but not the latter condition (i.e., desynchronization). In other words, neural networks for movement control were optimized, being regulated by smaller scale neural circuits. This view is in line with the overall cortical synchronization on the task observed on Type 1 and Type 4 performances and with the desynchronization observed on Type 2 and Type 3 performances. In particular, the higher ERD during Type 3 performance is evident in Fig. 2 (Theta ERD/ERS). On the other hand, theta ERS reflected the activation of neural networks involved in allocation of attention related to target stimuli rather than working memory (*Missonnier et al., 2006*). Finally, the ERS activity of Type 4 performance can be explained with the lower level of control of the action exerted by the participants during this type of suboptimal performance (see Table 1).

Moreover, we can hypothesize that the ERD/ERS pattern found in Type 1 performance is likely related to a "default mode network" functioning, proper to autonomous skills and goal-relevant attentional focus when approaching shot release (*Doppelmayr, Finkenzeller & Sauseng, 2008*), similar to what was found in golfers and basketball free throwers (*Chuang, Huang & Hung, 2013*; *Kao, Huang & Hung, 2013*). Excessive reinvestment in controlled processing undermines automaticity, and is related to higher cortical activity in the right parietal and frontal areas (*Broyd et al., 2009*), which is evident during Type 3 performance. Gentili and colleagues (*2011*) showed that with practice humans have the capacity to adapt their movements when challenged with novel demands or task requirements. They found a cortical adaptation of frontal temporal and parietal regions and a progressive idling of cortical rhythms in a learning paradigm of a visuo-motor task (*Gentili et al., 2011*). They suggested that the update of motor plans in response to incoming sensory information could be mirrored in theta band oscillation. There is also growing evidence that theta oscillations are linked to error monitoring (*Cavanagh, Cohen & Allen, 2009*; *Gevins & Smith, 2000*; *Luu, Tucker & Makeig, 2004*; *Trujillo & Allen, 2007*;

*Yordanova et al., 2004*). Error monitoring could explain the difference we obtained in the maps for optimal (Type 1 and 2) and suboptimal (Type 3 and 4) performance.

Findings in Alpha ERD/ERS (Figs. 3 and 4) concur with the evidence indicating relationships between: (1) low Alpha power and general cortical arousal, and (2) high Alpha power and task-relevant attentional processing (*Babiloni et al., 2008*; *Hatfield et al., 2013*; *Pfurtscheller & Lopes da Silva, 1999*). Indeed, we observed a significant difference among the different types of performance, particularly in FC2 site (see Supplemental Information), with ERD patterns (+1.12%) in low Alpha band for Type 3 performance. Thus, higher levels of general cortical arousal were associated with suboptimal performance states. In Type 2 performance cortical activity was synchronized on the task (−2.78%). ERS was more evident in Type 4 (−6.18%) and Type 1 performances (−9.14), indicating a low level of arousal and a minimal conscious control typical of flow-like behaviour (*Bortoli et al., 2012*; *Jackson & Csikszentmihalyi, 1999*). FC2 is one of the electrodes that represent the area where attentive processes and control on core components of action can be integrated for the control of goal-directed and stimulus-driven attention processes (*Corbetta & Shulman, 2002*).

Furthermore, the increase in the neuronal predictive power for action that was observed in the Fp1, F8, and P3 for high alpha band indicates that the optimization of neural networks can occur qualitatively in the form of enhanced proficiency of the neuronal representation of action. It is related to the visuo-spatial network that involves the switches between the performance types reflected in the activity of fronto-parietal areas during a few seconds preceding the shot release (i.e., Fpz, Fz, F8, Cp1; *Matsuzaka, Akiyama & Mushiake, 2013*). Moreover, the involvement of the de-synchronization of the contralateral sensory-motor area (i.e., C3) is an expression of the control of the right finger/hand (*Pfurtscheller & Lopes da Silva, 1999*). The results are also in agreement with the findings by Sauseng and colleagues (*2005*), who suggested that a shift of attention selectively modulates excitability of the contralateral posterior parietal cortex and that this posterior modulation of alpha activity is controlled by prefrontal regions.

In summary, the findings of the present study provided support to the construct of neural proficiency in sport performance that underlies Type 1 and Type 2 performance states as conceptualized in the MAP model (*Bortoli et al., 2012*), as well as in dual-process theories (*Furley, Schweizer & Bertrams, 2015*) and ACT (see *Wilson, 2008*). According to this view, neural proficiency is associated with Type 1 and Type 2 performance states, and mirrors the behaviour of elite athletes when they use efficient or effortful processing to deal with task demands. This is in agreement with the literature reviewed by *Klimesch (2012)*, who highlighted how alpha ERS and ERD reflect inhibition and release from inhibition, respectively. On the other hand, theta ERS reflects top-down processes related to working memory and attention (*Doppelmayr, Finkenzeller & Sauseng, 2008*).

From a neurophysiological perspective, successful goal-directed actions require the individuals to flexibly adapt their behaviour to deal with performance problems or environment changes, beyond correct action selection, planning, and execution (*Ullsperger, Danielmeier & Jocham, 2014*). Performance optimization is ensured by a continuous

monitoring of a wide network of brain areas (such as the posterior medial frontal cortex), which detects and evaluates deviations of actual from predicted states.

In conclusion, our findings provide evidence of distinct cortical processes underlying the four types of performance states derived from the interaction between performance and attention control levels. However, our methodological approach did not enable us to distinguish specific cortical networks among the four types of performance. Further investigation using the tools of graph theory (*Bullmore & Sporns, 2009*) and dynamic functional connectivity (*Hutchison et al., 2013*) within the framework of the MAP model, could provide a better understanding of the functional connectivity and interactions among default mode, dorsal and ventral attention network, sensory motor network, and visual network, typically involved in shooting performance. The investigation of ERD/ERS patterns after shot release could provide important information for a better understanding of the dynamics of brain oscillations during recovery from a highly demanding task execution.

From an applied point of view, our findings can address neurofeedback training to enhance performance in closed skill sports. Our findings can also help in choosing the proper areas to apply transcranial electrical stimulation (*Paulus, 2011*) in order to facilitate cortical plasticity, influencing the membrane potential of neurons and modulating spontaneous firing rates. Indeed, during sport performance, the brain must dynamically integrate, coordinate, and respond to internal and external stimuli across multiple time scales.

### Funding
The authors received no funding for this work.

### Competing Interests
Maurizio Bertollo and Silvia Comani are Academic Editors for PeerJ.

### Author Contributions
- Maurizio Bertollo conceived and designed the experiments, performed the experiments, analyzed the data, contributed reagents/materials/analysis tools, wrote the paper, prepared figures and/or tables, reviewed drafts of the paper.
- Selenia di Fronso performed the experiments, analyzed the data, wrote the paper, prepared figures and/or tables.
- Edson Filho reviewed drafts of the paper.
- Silvia Conforto and Maurizio Schmid analyzed the data, reviewed drafts of the paper.
- Laura Bortoli conceived and designed the experiments, performed the experiments, contributed reagents/materials/analysis tools.
- Silvia Comani conceived and designed the experiments, analyzed the data, contributed reagents/materials/analysis tools, prepared figures and/or tables, reviewed drafts of the paper.

- Claudio Robazza conceived and designed the experiments, performed the experiments, contributed reagents/materials/analysis tools, wrote the paper, prepared figures and/or tables, reviewed drafts of the paper.

## Human Ethics

The following information was supplied relating to ethical approvals (i.e., approving body and any reference numbers):

Comitato etico per la ricerca Biomedica, Università di Chieti-Pescara (ref. n. 10-21/05/2015).

## Data Availability

The raw data has been supplied as Data S1.

## Supplemental Information

Supplemental information for this article can be found online at http://dx.doi.org/10.7717/peerj.2082#supplemental-information.

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
