# Peer review of "Proficient brain for optimal performance: the MAP model perspective"

_PeerJ, doi:10.7717/peerj.2082_

## Round 0.1 · original submission · Minor Revisions

I believe both reviewers have provided useful comments for your effort to improve this manuscript. Please pay attention especially on the methodology and presentation of results to cearly present your work and findings.

Reviewer 1 ·

Basic reporting

The present study is theoretically sound and addresses well in the experimental design. The authors tried to validate the MAP model by adapting the EEG synchronization and desynchronization on Theta, low alpha, and high alpha power. The results are partially supportive of their claims. Overall, this paper was very well-conceived, conducted, and written and will make a significant contribution to the sport psychology literature. Several points are provided in an attempt to improve the paper and/or to provide suggestions for future research.

Experimental design

The main purpose of this study was to investigate the brain activation within the four types of performance. However, concerning the exact way to differentiate these four types of performance, seems not so evident and convinced. For example, page 13, line 325, the author explained the categorization of the four types of performance on EEG by using the shooting results and control levels. This statement seems not referable from previous contents. Moreover, how did the authors define the attentional control level? And how the attentional control level was recorded? It would be more proper that the authors could add an extra paragraph to address this categorization of EEG data.
Some minor comments regarding to the methodological section are listed below:

1. Page 12, line 288. Please specify the full name of the ISSF. And, it would be nice to indicate the years about how long the shooters have played.

2. Page 13, line 304. Please indicate the model name of the electronic scoring device.

3. Page 13, line 319. Did the microphone attach to the participants' body? If yes, it would be nice to let the readers know the exact position.

4. Page 14, line 350. What do you exactly mean "tshot" here?

5. Page 15, line 363. What did the authors exclude the "electrode" from the independent variables?

Validity of the findings

No Comments.

Additional comments

1. Page 22 line 467 TO 472, please rethink the citation regarding the relationship between anxiety and cortical activity here. In this study, the authors didn't measure the anxiety level during the shooting task. Also for the following inference of the relationship between anxiety and the cortical activity, authors should focus on the inference based on what they measured.

2. Page 23 line 493, what do you exactly differentiate the concept of the neural proficiency and the neural efficiency?

3. Page 24 line 516, the authors claimed that the performance are categorized by ERD and ERS. I'd like to suggest the authors that this inference should be cautious. Because the ERD/ERS can be indicated by different frequency bands. It depends on which frequency band that you want to compare with. The statement here seems too arbitrary.

4. Page 24 line 526 to 529. It would be nice that the authors can explain the possible causes regarding how the ERD activity is related to Type 3 performance.

5. Page 24 line 531 to 535. If the athletes are professional and perform the consistent shooting preparation from time to time, I would suggest to re-think the indication which states that the athletes use different strategies to maintain good performance.

Some minor suggestions:

1. Page 7 line 173, the reduced activation in the sensorimotor cortex, reflecting more efficient processing in motor performance. This work has also been done by Cheng et al. For example:
Sensorimotor rhythm neurofeedback enhances golf putting performance
&
Expert-novice differences in SMR activity during dart throwing

2. Page 7 line 178, the work by Babiloni should state clearly regarding on which ERD frequency band were measured. Otherwise, the readers could confuse about that.

3. Page 9 line 227, the statement of Type 4 performance is somehow inconsistent with automated state mentioned in previous text.

4. Page 9 line 232, the authors should try to explain the findings from Comani and di Fronso's work. How are the findings related to the MAP model could be the most important part of this argument.

5. Page 11 line 276, it would be nice to specify where are the areas which the authors mentioned about.

Reviewer 2 ·

Basic reporting

The article is written clearly and conforms to professional standards. It includes sufficient introduction and background. Relevant prior literature is appropriately referenced.
Figures are relevant to the content of the article but rather confusing and not sufficiently described.
The paper represents a proper 'unit of publication'.

Experimental design

The authors divide the eeg data in pre- and post-shot epochs but then only pre-shot results are presented. I think the post-shot analysis could be of interest, as well.

Validity of the findings

The main results in Figure 2, 3 and 4 are interesting but rather confusing: I assume that the three rows are top down (-1 to 0) s, (-2 to 1) s (-3 to 2) s. In this sense the caption and the notes on the figures are wrong. However, according to logic, the epoch with the least changes from the baseline should be the temporally closest to the baseline itself (-5 to -4) s. This makes the results difficult to interpret (possibly the time epochs are inverted in the notes on the figure, with respect to the topoplots but even the caption does not help in grasping this).

---

## Round 0.2 · Minor Revisions

The reviewer is now satisfied with the revision. The authors have my congratulations for addressing the point by point comments. However, I found that some minor issues regarding wording, typos, and referencing need to be addressed before I can accept your manuscript. Specific suggestions follow,

1. P7L178, please double check the citation of Cheng
et al., 2015, 2016, it looks to me the citation of the authors were not consistent for these two papers. In addition, the year of publication should be 2015 for both references.
2.Please check the spelling for Type 1 in figure 1
3.P15L371 there should be a period only in the end of the sentence.
4.Please align the number at Table 1
5.P21L447-448 it should be high alpha instead of theta. And the first sentence of the paragraph should be revised.

Please proofread the entire manuscript to ensure quality before resubmission.

Reviewer 2 ·

Basic reporting

I am now fine with the paper

Experimental design

no comments

Validity of the findings

no comments

---

## Round 0.3 · accepted · Accept

Dear Dr. Bertollo,

Thanks for your resubmission. After reading the revised manuscript, I've concluded that the manuscript can be accepted in its current condition.